# Composition of Eukaryotic Viruses and Bacteriophages in Individuals with Acute Gastroenteritis

**DOI:** 10.3390/v13122365

**Published:** 2021-11-25

**Authors:** Endrya do Socorro Fôro Ramos, Geovani de Oliveira Ribeiro, Fabiola Villanova, Flávio Augusto de Padua Milagres, Rafael Brustulin, Emerson Luiz Lima Araújo, Ramendra Pati Pandey, V. Samuel Raj, Xutao Deng, Eric Delwart, Adriana Luchs, Antonio Charlys da Costa, Élcio Leal

**Affiliations:** 1Laboratório de Diversidade Viral, Instituto de Ciências Biológicas, Universidade Federal do Pará, Belém 66075-000, Pará, Brazil; endrya.ramos@gmail.com (E.d.S.F.R.); geovanibiotec@gmail.com (G.d.O.R.); fvillanova@gmail.com (F.V.); 2Secretary of Health of Tocantins, Palmas 77453-000, Tocantins, Brazil; flaviomilagres@uft.edu.br (F.A.d.P.M.); eu3rafael@gmail.com (R.B.); 3Public Health Laboratory of Tocantins State (LACEN/TO), Palmas 77016-330, Tocantins, Brazil; 4General Coordination of Public Health, Laboratories of the Strategic Articulation, Department of the Health, Surveillance Secretariat, Ministry of Health (CGLAB/DAEVS/SVS-MS), Brasília 70719-040, Distrito Federal, Brazil; emerson.araujo@saude.gov.br; 5Centre for Drug Design Discovery and Development (C4D), SRM University, Delhi-NCR, Rajiv Gandhi Education City, Sonepat 131029, Haryana, India; ramendra.pandey@gmail.com (R.P.P.); deanacademic@srmuniversity.ac.in (V.S.R.); 6Vitalant Research Institute, 270 Masonic Avenue, San Francisco, CA 94143, USA; xdeng@vitalant.org (X.D.); Edelwart@Vitalant.org (E.D.); 7Department Laboratory Medicine, University of California San Francisco, San Francisco, CA 94143, USA; 8Virology Center, Enteric Disease Laboratory, Adolfo Lutz Institute, São Paulo 01246-000, São Paulo, Brazil; driluchs@gmail.com; 9Faculdade de Medicina, Instituto de Medicina Tropical, Universidade de São Paulo, São Paulo 05403-000, São Paulo, Brazil; charlysbr@yahoo.com.br

**Keywords:** gut virome, gastroenteritis, children, virus-like particles, viral diversity

## Abstract

Metagenomics based on the next-generation sequencing (NGS) technique is a target-independent assay that enables the simultaneous detection and genomic characterization of all viruses present in a sample. There is a limited amount of data about the virome of individuals with gastroenteritis (GI). In this study, the enteric virome of 250 individuals (92% were children under 5 years old) with GI living in the northeastern and northern regions of Brazil was characterized. Fecal samples were subjected to NGS, and the metagenomic analysis of virus-like particles (VLPs) identified 11 viral DNA families and 12 viral RNA families. As expected, the highest percentage of viral sequences detected were those commonly associated with GI, including rotavirus, adenovirus, norovirus (94.8%, 82% and 71.2%, respectively). The most common co-occurrences, in a single individual, were the combinations of rotavirus-adenovirus, rotavirus-norovirus, and norovirus-adenovirus (78%, 69%, and 62%, respectively). In the same way, common fecal-emerging human viruses were also detected, such as parechovirus, bocaporvirus, cosavirus, picobirnavirus, cardiovirus, salivirus, and Aichivirus. In addition, viruses that infect plants, nematodes, fungi, protists, animals, and arthropods could be identified. A large number of unclassified viral contigs were also identified. We show that the metagenomics approach is a powerful and promising tool for the detection and characterization of different viruses in clinical GI samples.

## 1. Introduction

The human virome refers to the composition of all eukaryotic and prokaryotic virus genomes that inhabit the human body [1,2]. Different studies have already demonstrated the composition of the virome in different conditions—for example, in diseases such as acute gastroenteritis (GI), which is an important public health problem in all age groups [3]. GI is estimated to kill over 500,000 people worldwide each year, and it is also the third leading cause of infant mortality, especially in low-income countries [4,5]. In Brazil, the incidence of infection is regular, being more frequent in the northeastern and northern regions, probably due to socioeconomic factors that are related to the mode of transmission of the disease (fecal–oral route) [6,7,8]. Gastrointestinal infection can be caused by a wide variety of microorganisms (viruses, bacteria, protozoa, helminths, and fungi). Viruses, on the other hand, are the most common, accounting for 70% of GI cases [5,6]. Norovirus (NoV), rotavirus (RV), enteric adenovirus (HAdV), sapovirus (SaV), and astrovirus (HAstV) are some of the viruses that have been reported in these instances [8,9,10,11,12].

Recently, after the implementation of the RV vaccine, other viral agents, such as human parechovirus (PeV), human bocaporvirus (HBoV), cosavirus (HCoSV), picobirnavirus (HPBV), cardiovirus B (Saffold virus—SAFV), saliviruses (SalV), and Aichiviruses (AiV), were described as possible agents associated with GI [13,14,15,16,17,18]. However, the contribution to these cases has not been fully elucidated. Moreover, some types of Enteroviruses were previously associated with GI of unknown etiology, but, because they are widely diverse, involved in a variety of clinical manifestations, and present tropism for the gastrointestinal tract [18], today, they are classically not associated with GI but are commonly identified in the human virome [19,20].

Although there is a diversity of diagnostic methods and studies, in many cases (approximately 40%), it is not possible to identify the etiology of the disease [8]. This, consequently, can lead to improper treatment and higher economic costs related to hospitalization and supplies used [21]. Such events may be associated with limitations in clinical and laboratory diagnosis, as well as the occurrence of emerging viruses or new viruses. This highlights the need for further research to elucidate these cases [7,15,22,23].

In this sense, next-generation sequencing (NGS) metagenomics is a powerful technique used to investigate the human viral composition in individuals with GI [24,25,26,27]. We employed next-generation sequencing (NGS) to isolate and purify virus-like particles (VLPs) from feces [28]. A study of VLPs in 250 individuals with gastroenteritis (GI) discovered a wide range of viruses in humans (RV, Nov, or HAdV). We also discovered a number of commensal viruses, unidentified viruses, and bacteriophages (phages).

## 2. Materials and Methods

### 2.1. Ethics Information

The survey was conducted in accordance with the 1975 Declaration of Helsinki (https://www.wma.net/what-we-do/medical-ethics-of-helsinki/ accessed on 10 March 2021), revised in 2013. The project was approved by the Committee of Ethics of the institutions involved (Faculty of Medicine, University of São Paulo (CAAE: 53153916.7.0000.0065), and Lutheran University Center of Palmas—ULBRA (CAAE: 53153916.7.3007.5516). There was no risk or harm to the children or their guardians; therefore, it was not necessary to apply the Informed Consent Term (ICF) in accordance with resolution 196/96 on research involving human beings—National Health Council (CNS)/Ministry of Health (MS), Brasília, 1996.

### 2.2. Study Population and Sample Collection

The current cross-sectional surveillance study was carried out in partnership with the Central Public Health Laboratory (LACEN) between 2010 and 2016 in the states of Tocantins (TO), Maranhão (MA), and Pará (PA), located in the north (TO/PA) and northeast (MA) of Brazil. Fecal samples were collected in 38 different municipalities (Figure 1). In this study, we analyzed 250 samples: 3 from Pará, 3 from Maranhão, and 244 from Tocantins. A total of 232 stool samples were collected from children under the age of five, three children between the ages of six and eleven, three adolescents between the ages of twelve and seventeen, one young person between the ages of eighteen and twenty-three, four adults between the ages of 24 and 59, and two elderly people over the age of sixty. Because of a lack of information, inadequate data, or the use of abbreviations to identify patient names, clinical data regarding age were not available for 5 patients. It is noteworthy that all patients on admission had GI symptoms (i.e., diarrhea, nausea, vomiting, and fever).

### 2.3. Sample Screening

The samples were initially delivered to LACEN/TO with an epidemiological investigation record that included the subjects’ demographic data (age, sex, and collection date) as well as clinical information (signs and symptoms). Subsequently, the samples were stored at −20 °C and forwarded to the USP Institute of Tropical Medicine of the University of São Paulo (IMT-USP) for identification of gastroenteric viruses (RV, NoV, HAdV, HAstV, HBoV, and SaV).

### 2.4. Viral-like Particle Metagenomics

The protocol used to perform the deep sequencing was a combination of several protocols applied to viral metagenomics and/or virus discovery, according to the procedures described previously [27,28,29,30,31]. Briefly, 50 mg of each human fecal sample was diluted in 500 μL of Hank’s Buffered Saline (HBSS) and added to a 2 mL impact-resistant tube containing lysis matrix C (MP Biomedicals, Santa Ana, CA, USA) and homogenized in a FastPrep-24 5G Homogenizer (MP biomedicals, USA). The homogenized sample was centrifuged at 12,000× *g* for 10 min and approximately 300 μL of the supernatant was percolated on a 0.45 μm filter (Merck Millipore, Billerica, MA, USA) to remove bacterial and eukaryotic cells. Approximately 100 μL of PEG-it virus precipitation solution (System Biosciences, Palo Alto, CA, USA) was added to the filtrate and the tube contents were gently homogenized, followed by incubation at 4 °C for 24 h. After the incubation period, the mixture was centrifuged at 10,000× *g* for 30 min at 4 °C and the supernatant (~350 μL) was discarded. The granulate, rich in virus-like particles (VLPs), was treated with a combination of nuclease enzymes (TURBO DNase and RNase Cocktail Enzyme Mix-Thermo Fischer Scientific, Waltham, MA, USA; Baseline ZERO DNase DNase-Epicenter, Madison, WI, USA; Benzonase-Darmstadt, Darmstadt, Germany, and RQ1 DNase-Free DNase and RNase A Solution-Promega, Madison, WI, USA) to digest unprotected nucleic acids. The resulting mixture was subsequently incubated at 37 °C for 2 h. Viral nucleic acids were then extracted using the ZR and ZR-96 viral DNA/RNA kits (Zymo Research, Irvine, CA, USA) according to the manufacturer’s instructions. cDNA synthesis was performed with an AMV reverse transcription reagent9 (Promega, Madison, WI, USA). Synthesis of second-strand cDNA was performed using a large fragment of DNA polymerase I (Klenow) (Promega). Subsequently, a Nextera XT Sample Preparation Kit (Illumina, San Diego, CA, USA) was used to build a DNA library, which was identified by a double barcode. The library was then purified using the ProNex^®^ size selective purification system (Promega, WI, USA). After ProNex^®^ purification, the amount of each sample was normalized to ensure equal representation of the library with the pooled samples using the ProNex^®^ NGS Library Quant Kit (Promega, WI, USA). For the size range, Pippin Prep (Sage Science, Inc., Beverly, MA, USA) was used to select a 300 bp tablet (range 200 to 400 bp), which excluded very short and long fragments from the library. Prior to cluster generation, libraries were quantified again by qPCR using the ProNex^®^ NGS Library Quant Kit (Promega, WI, USA). The library was sequenced in depth using a Hi-Seq 2500 Sequencer (Illumina, CA, USA) with 126 bp ends [23,32].

### 2.5. Bioinformatics Analysis

Deng et al. [33] previously described a protocol for bioinformatics analysis. Bowtie2 was used to eliminate non-viral sequences (i.e., human, bacterial, and fungal sequences). Only one random copy of duplicate reads was kept in read bases with a Phred quality score of less than 20. The Ensemble Assembler pipeline was used to assemble the remaining reads from scratch for each individual sample. BLASTx was used to link the contigs generated for eukaryotic and prokaryotic viral sequences to the RefSeq virus protein database. BLASTx was used to classify sequence hits with a lower E value (E value 0.001) to viruses than to non-viral proteins into their corresponding viral family and genus. Following virus identification, viral contigs were used to query (BLASTx E value < 1 × 10^−6^) the reference sequences, which were then used to map the entire genome using Geneious R9 software (Biomatters Ltd. L2, 18 Shortland Street, Auckland, 1010, New Zealand).

### 2.6. Statistics Analysis

All statistical tests were performed using JASP software v.0.11.1 (https://jasp-stats.org/ accessed on 6 September 2021).

## 3. Results

### 3.1. Characterization of Viruses in Individuals with GI

In order to explore the viral composition of individuals with GI, NGS was performed in 250 fecal samples obtained from different regions of Brazil (Figure 1). Among the participants, 92.80% (232/250) were children aged up to 5 years, 1.20% (3/250) were between 6 and 11 years old, 1.20% (3/250) between 12 and 17 years, 0.40% (1/250) between 18 and 23 years old, 1.60% (4/250) between 24 and 59 years old, and 0.80% (2/250) were elderly, aged 74 and 79 years old. In 2% (5/250) of patients, age information was not available. Among the patients, 59.20% (148/250) were male and 38.80% (97/250) were female. In 2% (5/250) of the samples, no clinical data were available. It is noteworthy that, at admission, all had symptoms of GI. Most patients experienced symptoms such as diarrhea, vomiting, and fever (Figure 1, upper right panel and Appendix A).

### 3.2. Viral Families Detected by NGS

In human fecal samples, bioinformatic analysis revealed a wide range of viruses. There were 11 DNA families discovered (*Mimiviridae*, *Nanoviridae*, *Adenoviridae*, *Anelloviridae*, *Baculoviridae*, *Circoviridae*, *Iridoviridae*, *Papillomaviridae*, *Parvoviridae*, *Phycodnaviridae*, and *Polyomaviridae*) and 12 RNA families (*Alphaflexiridae*, *Betaviridae*, *Astroviridae*, *Caliciviridae*, *Caliciviridae*, *Caliciviridae*, *Reoviridae*, *Rhabdoviridae*, *Secoviridae*, *Totiviridae*, and *Virgaviridaea*). They were classified as follows: (1) viruses identified to cause GI; (2) viruses from human hosts not connected to GI; (3) viruses from plant hosts; (4) viruses from nematodes, fungus, protozoa, and algae; (5) viruses from other animals and arthropod hosts; (6) viruses unclassified (new viruses); and (7) bacteriophages (phages) (Figure 2, Appendix A).

### 3.3. Pathogenic Viruses Associated with GI

Among the patients, the presence of at least one GI-related virus (RV, HadV, NoV, HastV, and SaV) was evidenced. Among these, 94.80% (237/250) were detected with RV, 82% (205/250) HAdV, 71.2% (178/250) NoV, 35.2% (88/250) HAstV, and 28% (70/250) SaV (Appendix A). The presence of two or more viruses in these patients was discovered, with double pathogens being more common (RV and HAdV in 195, RV and NoV in 172, and NoV and HAdV in 156) (Figure 3). Additionally, we also verified the identification of other viral agents commonly reported in association with GI cases as PeV in 11.20% (28/250), HBoV in 2% (10/250), HCoSV in 2% (5/250), HPBV 1.60% (4/250), SalV in 1.60% (4/250), SAFV 1.20% (3/250), and AiV in 0.40% (1/250). Previous studies [13,14,16,17,34,35,36,37] have provided a detailed description of the clinical data as well as genome characterization of the GI viruses discovered here.

### 3.4. Other Viruses in Human Hosts Not Related to GI

Among viral sequences, we also identified other viruses that infect human hosts, with a predominance of the *Picornaviridae* family, the Enterovirus genus in 37.60% (94/250) of the samples (Figure 2), attributed to Enterovirus species A-D and J in 37.20% (93/250), and Rhinovirus A–C in 5.20% (13/250). It is noteworthy that these agents are involved in a variety of clinical conditions, being commonly identified in the gastrointestinal tracts of symptomatic or asymptomatic patients.

We also identified other families, such as *Parvoviridae* and *Reoviridae*. These viral species (adeno-associated dependoparvovirus A, Anseriform dependoparvovirus, Deltapapillomavirus 4, human polyomavirus 10, mammalian orthoreovirus) infect humans and have no direct relationship with cases of GI. In previous studies [38,39,40,41,42], the genome characterization of public health interest viruses has been described [38,39,40,41,42].

### 3.5. Viruses from Different Hosts

In the same way, the presence of viral families (*Mimiviridae*, *Betaflexiviridae*, *Alphaflexiviridae*, *Anelloviridae*, *Endornaviridae*, *Virgaviridae*, *Baculoviridae*, *Cyclovirus*, *Totiviridae*, *Secoviridae*, *Anelloviridae*, *Nanoviridae*, *Iridoviridae*), which are not linked with human infection, was also found (Figure 2). Among these viruses, some infect plants, such as squash mosaic virus, identified in 19.60% (49/250) of patients; Cucumis melo alphaendornavirus in 1.60% (4/250) [23]; potato virus M in 0.80% (2/250), apple stem grooving virus, cactus virus X, pepper mild mottle virus, potato virus M, shallot latent virus, and faba bean yellow leaf virus in 0.40% (1/250) each. Nematode, fungi, and protist viruses were also seen in some samples: Xinzhou alphanemrhavirus in 1.20% (3/250), Saccharomyces cerevisiae virus LA, Cafeteria roenbergensis virus, Acanthamoeba polyphaga mimivirus, Micromonas pusilla virus SP1, and Paramecium bursaria Chlorella virus 1 in 0.40% each (1/250). Other animals and arthropod viruses were detected in many samples: torque teno virus at 21.60% (54/250), rodent-associated cyclovirus and Invertebrate iridescent virus at 0.80% (2/250) each, torque teno virus-like mini virus, chicken anemia virus, Chimpanzee-associated circovirus 1, and Plodia granulovirus interpunctella in 0.40% (1/250) each (Appendix A).

### 3.6. Unclassified Viruses

In addition, viruses not assigned to a classification (Rctr197k virus, Wuhan large pig roundworm virus 1 [43], Xinzhou partiti-like virus 1, Rhizoctonia solani mitovirus 6, Organic Lake phycodnavirus, Jingmen toti-like virus 1, Human fecal virus Jorvi2, Husavirus [44], Indivirus, Hubei partiti-like virus 55, Hubei toti-like virus 12, Hudisavirus, Giant panda associated partiti-like virus, Gokushovirus, Halovirus, Columbid circovirus, Chimpanzee anellovirus, Chrysochromulina ericina virus) (Appendix A) were also found.

### 3.7. Bacteriophages

Furthermore, bacteriophages were detected in a large proportion of patients (Figure 2). In summary, Escherichia virus spp., Lactococcus virus spp., and Streptococcus virus spp. were the most detected in the samples, with frequencies of 15.20% (38/250), 7.60% (19/250), and 7.20% (18/250), respectively (Appendix A).

## 4. Discussion

The human gut virome is formed by a variety of species, especially phages and RNA viruses [3,8,27,45,46,47]. Advances in technology aimed at virus detection, such as the NGS-based metagenomics approach, have enabled numerous contributions, among which the investigation of the viral community stands out, which can help to elucidate the etiology of various infectious diseases, especially in cases where there is no knowledge about the causative agent [3,8,22,27,48,49].

In this study, NGS was performed on 250 fecal samples from individuals with GI. Overall, 92% of the samples were from children under 5 years old. The sequencing resulted in the identification of a great variety of viruses. We were able to assign reads to 11 DNA and 12 RNA families. The other reads were related to bacteriophages and new, unclassified viruses. Other studies have also investigated the intestinal virome in these patients. However, studies of the virome in GI cases are still limited [3,22,27].

RV, NoV, HAdV, HAstV, and SaV are well-known and recognized as important GI causative agents [50]; therefore, finding them at a high frequency in the present study is not a surprise. NGS identified RV (94.80%) as the most prevalent viral agent in these GI cases. This result was expected, as RV is the leading cause associated with viral GI in young children worldwide, accounting for nearly 125,000 deaths annually [4]. HAdV was the second most detected viral agent in the samples (82%). HAdV is widely reported in GI cases [8,51], but it has also been associated with a spectrum of clinical presentations, including respiratory, neurological, ocular, and urinary infections [52]. NoV was also detected at a high frequency (71.20%) in our samples. NoVs are highly transmissible viruses that cause acute gastroenteritis worldwide, and they are recognized as the leading cause of epidemic and sporadic GI in children and adults [53]. Additionally, HAstV and SaV were also commonly detected in our samples, in 35.46% and 28%, respectively. Both of them have been more recently associated with GI cases [31,54]. We also found samples in which multiple pathogenic viruses were identified by NGS. Notably, the most common occurrences were the combinations of RV-HAdV, RV-NoV, and NoV-HAdV (78%, 69%, and 62%, respectively). Other studies have reported the presence of multiple viruses in individuals with GI [3,8,55,56].

In addition, we found other viral agents that were recently described in cases of GI [3]. PeV, HBoV, HCoSV, HPBV, SAFV, SalV, and AiV were detected in 11.20%, 2%, 2%, 1.60%, 1.20%, 1.60%, and 0.40% of samples [11,12,13,14,15,16,17,18]. Emerging human picornaviruses, including PeV, AiV, and SalV, were found to be associated with GI, but their roles in enteric infections are not fully understood [57]. In the same way, the pathogenicity of HCoSV, another member of the *Picornaviridae* family, remains unclear [58]. HBoV is a newly identified human parvovirus linked to respiratory and gastrointestinal diseases [59]. Saffold virus is a novel human cardiovirus that was identified in 2007, and its pathogenicity and clinical characteristics have not been elucidated [60]. Picobirnaviruses are ubiquitous in the feces and gut contents of humans and other animals with or without diarrhea. They are considered opportunistic enteric pathogens of mammals and avian species [61]. Some members of the Enterovirus genus (Coxsakievirus, echovirus, enterovirus C99, and enterovirus B73) were also detected in 37.60% of samples. Enteroviruses have been associated with a great variety of clinical manifestations but have also been detected in asymptomatic individuals [62,63]. A large number of newly described viruses were identified in the present investigation, including mammalian orthoreovirus [41], Wuhan large pig roundworm virus [43], and husavirus [44]. These findings suggest that these new viral species are circulating in the human population, highlighting the urgent need for a more robust surveillance system that focuses on detecting and molecularly characterizing newly described fecal viruses in order to improve our understanding of viral diversity and be ready to respond quickly in the event of emerging pathogen outbreaks.

We also observed viruses that infect plants, nematodes, fungi, protists, animals, and arthropods. Several studies conducted in different countries, including Brazil, demonstrate the presence of these viruses in human fecal content, arising likely from human contact with the environment, water and food [23,64,65]. It is worth noting that human contact with wild animals may favor spillover events or even recombination events, making it important to investigate the circulation of these viruses as a health prevention measure. An important feature of virome analysis is that the epidemiological background information on potential contact with animals, consumption of contaminated food, and/or medical records of the patients is not generally available. The present study is no exception; there was no epidemiological connection between the patient and the virus. Fecal samples have significant dietary content, and plant, fungi, and protist viral nucleic acid are expected to appear in large amounts in the resulting sequences [66]. Considering animal, arthropod, or nematode sequences, the patient could have been previously infected or had contact with these viral agents. In addition, Gyrovirus 1, Alphatorquevirus 53, and Betatorquevirus 1 were detected in the samples. *Anelloviridae* are extremely diverse and can be found in many human body sites in a large fraction of all humans examined, and no specific pathogenic effects have been linked to *Anelloviridae* until now [63]. Greater abundance of *Anelloviridae* has been found in individuals who are immunocompromised, including the recipients of lung transplants, individuals who are HIV-positive, and individuals on immunosuppressive medications owing to inflammatory bowel disease, indicating that *Anelloviridae* are normally under host immune control [67].

Furthermore, sequences from phages were identified in the samples. It is recognized that phages are the most abundant viruses within the viral community, as shown by the genome sequences detected among the samples [1,2,68]. The human gastrointestinal tract is inhabited by a complex system of bacteria, viruses, fungi, archaea, protists, and eukaryotic parasites, with a predominance of bacteria and bacterial viruses (bacteriophages). Collectively, these microbes form the microbiota of the microecosystem of humans [69]; therefore, one would expect to find a high frequency of these agents in human fecal material.

In conclusion, our findings highlight the importance of investigating GI cases of unknown etiology in order to better understand the molecular and epidemiological aspects that may be involved in the development of the infection. At the same time, such knowledge can contribute to our understanding of viral diversity, which is still poorly explored. These data confirm the importance of new diagnostic approaches and lend support to the efficacy of the metagenomic tools that aim to investigate GI cases with unknown etiology [3,8,27,36,48,49]. We infer that many GI cases of unknown etiology may be related to routinely tested viruses, but that the diagnostic methods used do not allow us to identify them. This can happen because most of the tests in use in the laboratory routine are directed at specific, well-characterized genotypes or are dependent on specific sequences of previously known viral strains [3,22]. Consequently, in the case of unusual genotypes, mutations and rearrangements can cause low efficiency in viral detection [3,22].

## Figures and Tables

**Figure 1 viruses-13-02365-f001:**
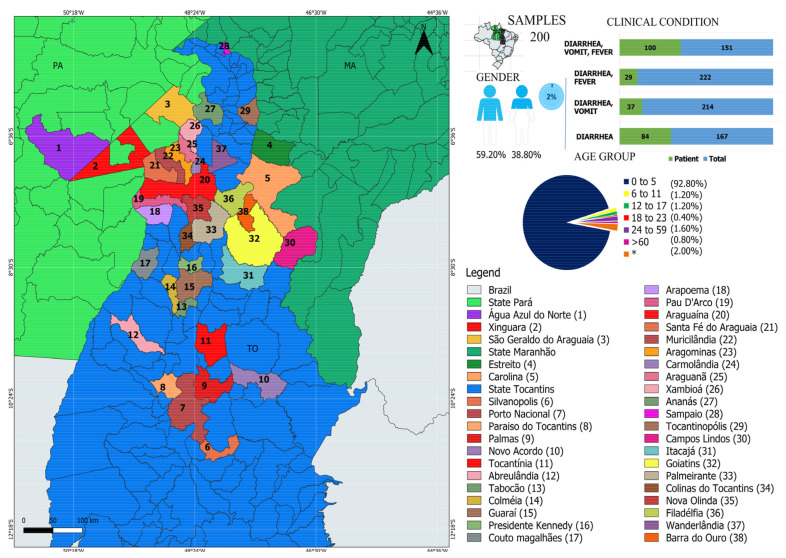
The municipalities where samples were gathered are shown on the map. The municipalities where samples were collected are highlighted on the map by numbers. According to the map, the legend in the lower right panel indicates the municipalities with distinct colors. TO denotes the state of Tocantins, MA denotes the state of Maranho, and PA denotes the state of Pará. The proportions of participants by gender and clinical symptoms are shown in the upper right panel. The map was created using the QGIS Geographic Information System software (https://www.qgis.org/ accessed on 6 September 2021) and data from the IBGE (https://www.ibge.gov.br/ accessed on 6 September 2021).

**Figure 2 viruses-13-02365-f002:**
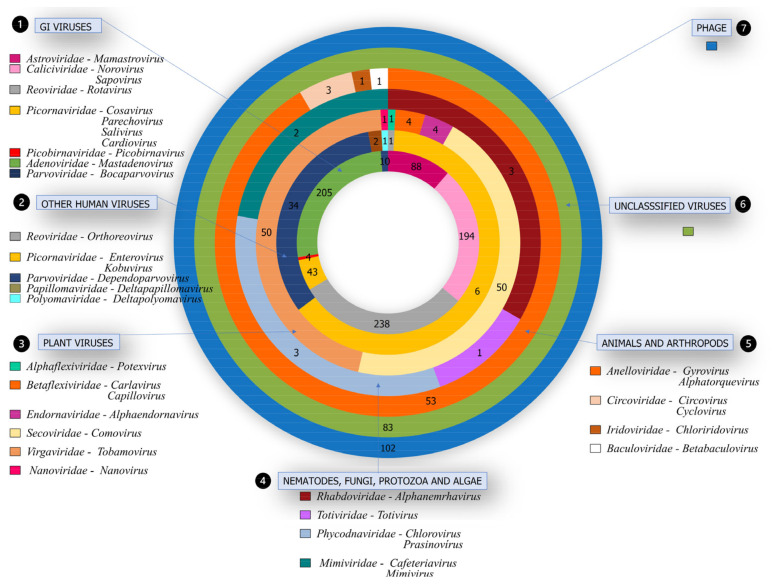
Representation of the number of viral families, according to gender, identified in the 250 cases of gastroenteritis of unknown etiology. The seven rings highlight the number of patients per family. The rings demonstrate the organization into five groups: 1—viruses causing or related to gastroenteritis, 2—other viruses that infect humans without direct relation to GI, 3—viruses from different hosts, 4—number of samples in which viruses were newly detected, 5—number of samples that were detected in viral sequences from phages.

**Figure 3 viruses-13-02365-f003:**
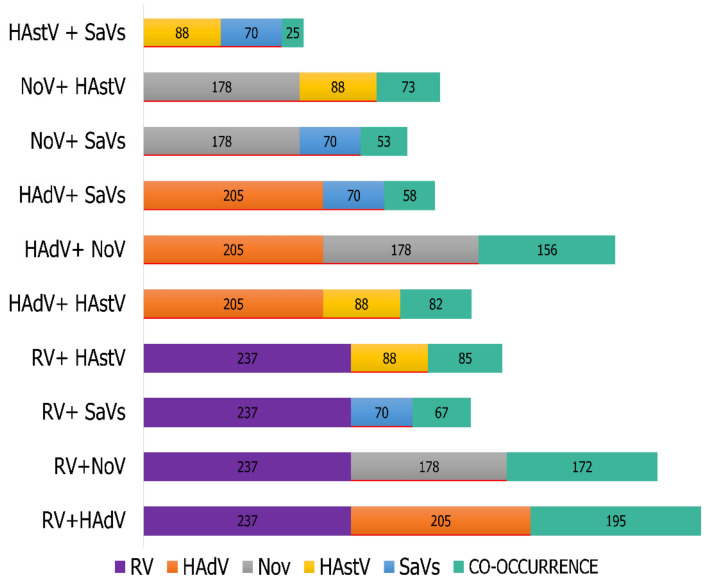
Distribution of rotavirus (RV), adenovirus (HAdV), norovirus (NoV), sapovirus (SaV), and astrovirus (HAstV). Highlighted in dark blue are 237 cases of RV detection; in orange, 205 of HAdV; in gray, 178 of NoV; in yellow, 88 of HAstV; in light blue, 70 of SaV; and in green, the occurrence of concomitant viruses in the samples.

## Data Availability

Not applicable.

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
