# Peer review of "Composition of Eukaryotic Viruses and Bacteriophages in Individuals with Acute Gastroenteritis"

_viruses, 2021, doi:10.3390/v13122365_

Round 1

Reviewer 1 Report

The authors used NGS with an approach aimed to isolate and purify virus-like particles (VLPs) from fecal samples, classically associated with GI. This is a well conducted scientific research. However, the authors can revise the use of their punctuations throughout the article. There are lots of misused grammar which could be misleading. Also, I think the article could benefit from English editing by a native english speaker.

Author Response

All your comments were addressed accordingly in this new version of the manuscript.

Reviewer 2 Report

Major concerns:

  1. Line 105: It is mentioned that “It is noteworthy that all patients on admission will have GI symptoms”. What it is mean will have symptoms? There were no GI symptoms when the fecal sample was collected? How long it taken to show the GI symptoms after sample collection?
  2. Lines 95-105: No control sample from the healthy people with the similar age?

    Since the number of the samples from the patients elder than 17-year-old are very limited, like totally 7, why just exclude them from the cohort?

  1. Lines 126-134: The supernatant (~300μl) was collected from the homogenized sample after centrifuging at 12,000 ×g. Then, it was mixed with precipitation solution (~100 μl) and centrifuged at 10,000 ×g after incubation to get the pellet which rich in virus like particles. Why centrifuging the homogenized sample at higher speed to remove the debris than centrifuging the sample to get the VLPs? How it possible the VLPs will not spin down at the first high-speed centrifuge?
  2. Nowadays, the shotgun metagenomic sequencing technology is very popular to study the gut microbial components, including virus and bacteria, and their functions. Is there special reason why the authors choose the method using this study, not the shotgun sequencing method except the financial concerns?
  3. Line 187-189: More details should be provided here, such as what statistic method used in this study.
  4. What about the sequencing counts of the virus from different sample? Are there quite different between samples or very similar?

Minor concerns:

  1. Introduction: Paragraph 1 (lines 47-51) and paragraph 2 (lines 52-56) may consider combine to one paragraph which describe the gastroenteritis diseases.

Similarly, Paragraph 3 (lines 57-61) and paragraph 4 (lines 62-70) may combine to one to describe the microbiome and pathogens for gastroenteritis.

  1. Lines 195-202: Please correctly use the dot “.” and comma “,” in the numbers. It should be “92.8%” not “92,8%”, there are totally different. Please also go through the whole manuscript and correct it.
  2. Lines 298-302: Some background of these bacteriophages may be briefly described here.

Author Response

Major concerns:

Line 105: It is mentioned that “It is noteworthy that all patients on admission will have GI symptoms”. What it is mean will have symptoms? There were no GI symptoms when the fecal sample was collected? How long it taken to show the GI symptoms after sample collection?

Resp: All individuals presented acute GI when fecal samples were collected. After medication, all individuals recovered from GI symptoms. We have no information on how long the symptoms have lasted.

Lines 95-105: No control sample from the healthy people with the similar age?

Since the number of the samples from the patients elder than 17-year-old are very limited, like totally 7, why just exclude them from the cohort?

Resp: We decided to keep these individuals (> 17-Year-old) for convenience and because their data had no negative impact on the results. Because this is a purely descriptive study we haven't control (GI negative) samples.

Lines 126-134: The supernatant (~300μl) was collected from the homogenized sample after centrifuging at 12,000 ×g. Then, it was mixed with precipitation solution (~100 μl) and centrifuged at 10,000 ×g after incubation to get the pellet which rich in virus like particles. Why centrifuging the homogenized sample at higher speed to remove the debris than centrifuging the sample to get the VLPs? How it possible the VLPs will not spin down at the first high-speed centrifuge?

Resp: This technique is used to enrich virions outside cells, in the first centrifugation “heave” particles such as cells will precipitate, and VLPs remained in the supernatant.

Nowadays, the shotgun metagenomic sequencing technology is very popular to study the gut microbial components, including virus and bacteria, and their functions. Is there special reason why the authors choose the method using this study, not the shotgun sequencing method except the financial concerns?

Resp: We have selected the current methodology of VLP enrichment and bioinformatics analysis based on previous validations of this approach (see these studies for example; 10.1038/s41467-018-06502-9; 10.1186/s12864-014-1207-4).

Line 187-189: More details should be provided here, such as what statistic method used in this study.

What about the sequencing counts of the virus from different sample? Are there quite different between samples or very similar?

Resp: Most reads detected in all individuals were those related to viruses associated with GI. No significant differences were observed regarding the number of viral reads among the individuals. We have included information on sequencing and blast search of each individual in the supplementary data.

Minor concerns:

Introduction: Paragraph 1 (lines 47-51) and paragraph 2 (lines 52-56) may consider combine to one paragraph which describe the gastroenteritis diseases.

Similarly, Paragraph 3 (lines 57-61) and paragraph 4 (lines 62-70) may combine to one to describe the microbiome and pathogens for gastroenteritis.

Resp: We have changed these sentences accordingly.

Lines 195-202: Please correctly use the dot “.” and comma “,” in the numbers. It should be “92.8%” not “92,8%”, there are totally different. Please also go through the whole manuscript and correct it.

Lines 298-302: Some background of these bacteriophages may be briefly described here.

Resp: We have modified this kind of error in the manuscript

Reviewer 3 Report

Major points:

This study investigated the enteric viral composition of 250 individuals with acute gastroenteritis in Brazil. Due to lack of relevant data released in Brazil, the value of this manuscript was significant. This manuscript majorly described the viral composition of several NGS data and reveal the circulating virus which could cause the acute gastroenteritis in individuals.

However, this manuscript only describes the demographic characteristics, geographic distribution and viral detection situation, and it lack the deep analysis about the viral genomes associated with acute gastroenteritis in human beings (e.g., rotavirus, norovirus and sapovirus). Although the authors have assembled a lot of genomes from NGS data in this study and identified exist of infection or co-infection with acute gastroenteritis, but the genomic length of these viruses (rotavirus, norovirus, sapovirus and others) were not shown. If the viral genomic length was short (e.g., the coverage of norovirus was shorter than 50% of full-length), the genome might not be used as a positive result. I suspect that the genomic length of these viruses are not full-length (that mean a low abundance of virus) so that the infection could not be identified. Actually, the authors did not use other methods (e.g., PCR, qPCR) to confirm the viral exist in samples. So, we did not agree with the number results of infection or co-infection only through the assembled genomes. Please add according results in this manuscript and explain it.

Additionally, previous studies have confirmed the molecular epidemiological characteristics of rotavirus, norovirus and sapovirus, but this manuscript did not extend the research conclusion. Although the authors described a large of viral number in this manuscript, but they did not majorly explain the pathogens that caused acute gastroenteritis, which was an important point for this study. We suggest that the authors should add the according content in this manuscript, especially the viruses related to acute gastroenteritis (e.g., add the molecular epidemiological analysis of rotavirus, norovirus and sapovirus in a new section).

Moreover, this manuscript lacks the NGS output data summary in the section 3.1, and it did not clarify the samples distribution during the NGS library construction (e.g., one sample for one library?). The authors should clarify these results in manuscript.

The English expression in this manuscript must be improved, e.g., “Highlighting Norovirus (NoV), Rotavirus (RV), Enteric Adenovirus (HAdV), Sapovirus (SaV) and Astrovirus (HAstV) as they are commonly reported in these cases. (Page 2, Line 59-60)” and “Yet some types of Enteroviruses were previously associated with GI of unknown etiology, but because they are widely diverse, involved in a variety of clinical manifestations and present tropism for the gastrointestinal tract [18], today are classically not associated with GI, but commonly identified in human virome. (Page 2, Line 66-69)”. Similar grammar mistakes in other paragraphs are observed (Page 6, Line 233-247). We raise several grammar mistakes during the peer review. We don’t suggest the authors to use a complex sentence structure for clarifying the scientific results. And the literary words should be great for clarifying the opinion, because it could improve the readability for all readers.  

Minor points:

Page 1, Line 34, “The metagenomic study of viruses identified from virus-like particles (VLPs) recovered from fecal samples detected reads belonging to 11 viral DNA families and 12 viral RNA families”. Please rephrase this sentence. We suggest to use the attributive clause to avoid confusion.

Page 2, Line 52-53, “Worldwide, it is estimated that there are > 500,000 deaths per year from the disease and it is also considered the third cause of infant mortality, especially in low-income countries” The authors should point the accurate types of diseases which cause > 500,000 deaths per year, even though I could sense that it was acute gastroenteritis.

Page 2, Line 58, “However, viruses are the most prevalent, corresponding to 70% of GI cases [5,6]” Please modify this sentence to “However, viruses are the most important pathogens, corresponding to 70% of GI cases.”

Page 2, Line 66-69, “Yet some types of Enteroviruses were previously associated with GI of unknown etiology, but because they are widely diverse, involved in a variety of clinical manifestations and present tropism for the gastrointestinal tract [18], today are classically not associated with GI, but commonly identified in human virome.” Please rephrase this sentence because it contains the grammar mistake.

Page 2, Line 79, “Here we used NGS with an approach aimed to isolate and purify virus-like particles (VLPs) from fecal samples” The NGS could not isolate and purify virus-like particles (VLPs) from fecal samples. The samples process procedure was performed to complete this aim. Please rephrase this sentence.

Page 2, Line 79, “The study of VLPs in 250 individuals with gastroenteritis (GI), identified a variety of viruses including GI in humans (RV, Nov or HAdV).” Please remove the word “including GI” in this sentence.

Page 3, Line 103, “between 18 and 23 years, 4 adults aged 24 to 59 years and 2 elderly > 60 In 5 patients” Please add the period between “elderly > 60” and “In 5 patients”.

Page 3, Line 105, “It is noteworthy that all patients on admission will have GI symptoms.” The authors need to add the description or definition about clinical symptoms of acute gastroenteritis.

Page 4, Line 177, “The cleaned reads were first assembled individual DBG assemblers, partitioned assemblers or Mira4” Please rephrase this sentence.

Page 4, Line 158-179 The authors conducted two different principle-based assemblers (DBG assemblers and CL assemblers) to harvest the longer genomes. However, the assembly procedure described in this manuscript was chaotic so that the readers could not clearly catch the significant points. We suggest that the authors should simplify the description about assembly procedure.

Page 4, Line 195, “92,80% (232/250) were children” The comma symbol was incorrect. Please correct this error at several position throughout this manuscript.  

Page 5, Line 213-220, The authors implemented two attributive clauses (among which) in this paragraph. The content is right for describing the viral taxonomy, but the grammar structures need to be modified.

Page 6, Line 236-238, “In these patients, the occurrence of coinfection was observed. being more common between double infection, RV and HAdV in 196, RV and NoV in 173 and NoV and HAdV in 157. triple infection was verified being more frequent among RV, HAdV and NoV in 60,80% (152/250) of patients.” Please correct the grammar mistakes. And the number of co-infection is not consistent with the results of Figure 3. Please carefully check and illustrate the phenomenon.

Page 8, Line 294-295, “Detailed descriptions of the characterization of some of the viruses detected in these samples have been previously reported.” Please rephrase this sentence.

Page 9, Line 340-341, “This finding indicate that more than one virus can trigger the disease or even contribute to clinical severity.” This study did not summarize the relationship between clinical severity and co-infection and did not perform the statistics. This sentence is not precise, please re-organized.

Page 9, Line 352-353, “Our study indicates that although not related with gastroenteritis enteroviruses were commonly found in association with other pathogenic viruses.” Grammar logic mistake.

Page 9, Line 357, “Another interesting finding identified in the samples from these patients.” Grammar mistakes.

Page 9, Line 358, “This fact highlights the circulation of new viral types among the population,” The authors found a large number of new virus sequences not assigned to classification, but they thought that these genomic sequences belonged to new viral types of circulation. Although these viral sequences were not assigned to classification, it could be the reason that the viral database lack the according viral taxonomy. We don’t not agree with authors that the circulation of new viral types among the population. If the authors want to illustrate the ideas, they need to perform a systemic and deep studies focusing on these new viral types.

Figure 1, “Map showing the location of the states (SP, PA, MA and TO) and municipalities in which the 251 samples” Please modify the grammar mistake. And the dpi of all figures in production need be improved.

Figure 2, The authors declare 251 cases in the legend, resulting in contradiction with previous content. “OUTHER VIRUS HUMAN” The figure contains a word grammar mistake.

Figure 3, We suggest that the authors could draw another graphic (single bar plot) to show the number of co-infection within the Figure 3. Because the current graphic plot of graphic cause confusion about co-infection and single infection so that we could not easily identify the number of single infection (e.g., the number of RV).

Table S2. We did not find the Table S2 in the supplementary materials.

Author Response

Major points:

This study investigated the enteric viral composition of 250 individuals with acute gastroenteritis in Brazil. Due to lack of relevant data released in Brazil, the value of this manuscript was significant. This manuscript majorly described the viral composition of several NGS data and reveal the circulating virus which could cause the acute gastroenteritis in individuals.

However, this manuscript only describes the demographic characteristics, geographic distribution and viral detection situation, and it lack the deep analysis about the viral genomes associated with acute gastroenteritis in human beings (e.g., rotavirus, norovirus and sapovirus).

Resp: We appreciate this comment. We agree that this manuscript is purely descriptive and is incomplete regarding the detailed characterization of important viruses found in the samples. The current manuscript provides a summary of all the reads detected by NGS. Our group has previously performed and published a detailed characterization of the majority of viruses found in these GI patients.(Viruses. 2018 Dec 28;11(1):16. doi: 10.3390/v11010016. Viruses. 2018 Sep 24;10(10):520. doi: 10.3390/v10100520. Virus Genes. 2018 Jun;54(3):470-473. doi: 10.1007/s11262-018-1557-0. Epub 2018 Mar 28 Sci Rep. 2018 Aug 17;8(1):12304. doi: 10.1038/s41598-018-30214-1. Viruses. 2019 May 29;11(6):488. doi: 10.3390/v11060488. Mem Inst Oswaldo Cruz. 2019 Apr 4;114:e180574. doi: 10.1590/0074-02760180574. Sci Rep. 2019 Dec 9;9(1):18599. doi: 10.1038/s41598-019-55216-5. Viruses. 2020 May 4;12(5):508. doi: 10.3390/v12050508. J Gen Virol. 2020 Dec;101(12):1280-1288. doi: 10.1099/jgv.0.001500. Epub 2020 Oct 12. Arch Virol. 2021 Mar;166(3):905-913. doi: 10.1007/s00705-020-04944-5. Epub 2021 Jan 19. Viruses. 2021 Mar 31;13(4):595. doi:10.3390/v13040595. PLoS One. 2021 Mar 23;16(3):e0248486. doi: 10.1371/journal.pone.0248486. eCollection 2021. Arch Virol. 2021 Nov;166(11):3165-3172. doi: 10.1007/s00705-021-05206-8. Epub 2021 Aug 21. Microorganisms. 2021 Jun 22;9(7):1349. doi: 10.3390/microorganisms9071349.).

The results were confirmed using PCR and Sanger sequencing. Although some of these articles were addressed in this study, most were not included in the current manuscript's references to avoid excessive self-citation.

Although the authors have assembled a lot of genomes from NGS data in this study and identified exist of infection or co-infection with acute gastroenteritis, but the genomic length of these viruses (rotavirus, norovirus, sapovirus and others) were not shown. If the viral genomic length was short (e.g., the coverage of norovirus was shorter than 50% of full-length), the genome might not be used as a positive result. I suspect that the genomic length of these viruses are not full-length (that mean a low abundance of virus) so that the infection could not be identified. Actually, the authors did not use other methods (e.g., PCR, qPCR) to confirm the viral exist in samples. So, we did not agree with the number results of infection or co-infection only through the assembled genomes. Please add according results in this manuscript and explain it.

Resp: We completely agree that the detection of viral readings does not always signify infection. Although this is only descriptive research, prior investigations have retrieved full-length sequences and thorough descriptions of viruses linked to the gastrointestinal tract. Indeed, we performed PCR and Sanger sequencing to confirm some of the findings. We have included information on the read sizes, e-values, and read mapping to virus species for each person in the supplementary materials.

Additionally, previous studies have confirmed the molecular epidemiological characteristics of rotavirus, norovirus and sapovirus, but this manuscript did not extend the research conclusion. Although the authors described a large of viral number in this manuscript, but they did not majorly explain the pathogens that caused acute gastroenteritis, which was an important point for this study. We suggest that the authors should add the according content in this manuscript, especially the viruses related to acute gastroenteritis (e.g., add the molecular epidemiological analysis of rotavirus, norovirus and sapovirus in a new section).

Resp: We have changed the discussion accordingly

Moreover, this manuscript lacks the NGS output data summary in the section 3.1, and it did not clarify the samples distribution during the NGS library construction (e.g., one sample for one library?). The authors should clarify these results in manuscript.

Resp: We have included data of each individual corresponding to the sequencing and the results of blast searches in the supplementary material.

The English expression in this manuscript must be improved, e.g., “Highlighting Norovirus (NoV), Rotavirus (RV), Enteric Adenovirus (HAdV), Sapovirus (SaV) and Astrovirus (HAstV) as they are commonly reported in these cases. (Page 2, Line 59-60)” and “Yet some types of Enteroviruses were previously associated with GI of unknown etiology, but because they are widely diverse, involved in a variety of clinical manifestations and present tropism for the gastrointestinal tract [18], today are classically not associated with GI, but commonly identified in human virome. (Page 2, Line 66-69)”. Similar grammar mistakes in other paragraphs are observed (Page 6, Line 233-247). We raise several grammar mistakes during the peer review. We don’t suggest the authors to use a complex sentence structure for clarifying the scientific results. And the literary words should be great for clarifying the opinion, because it could improve the readability for all readers.  

Resp: We apologize for the poor quality of language used in our manuscript; in the updated version, we attempted to clarify the content by using more direct and technical sentences.

Minor points:

Page 1, Line 34, “The metagenomic study of viruses identified from virus-like particles (VLPs) recovered from fecal samples detected reads belonging to 11 viral DNA families and 12 viral RNA families”Please rephrase this sentence. We suggest to use the attributive clause to avoid confusion.

Resp: We have modified this sentence

Page 2, Line 52-53, “Worldwide, it is estimated that there are > 500,000 deaths per year from the disease and it is also considered the third cause of infant mortality, especially in low-income countries” The authors should point the accurate types of diseases which cause > 500,000 deaths per year, even though I could sense that it was acute gastroenteritis.

Resp: We have modified this sentence.

Page 2, Line 58, “However, viruses are the most prevalent, corresponding to 70% of GI cases [5,6]” Please modify this sentence to “However, viruses are the most important pathogens, corresponding to 70% of GI cases.

Resp: This sentence was changed in the manuscript.

Page 2, Line 66-69, “Yet some types of Enteroviruses were previously associated with GI of unknown etiology, but because they are widely diverse, involved in a variety of clinical manifestations and present tropism for the gastrointestinal tract [18], today are classically not associated with GI, but commonly identified in human virome.” Please rephrase this sentence because it contains the grammar mistake.

Resp: We have modified this sentence in the manuscript.

Page 2, Line 79, “Here we used NGS with an approach aimed to isolate and purify virus-like particles (VLPs) from fecal samples” The NGS could not isolate and purify virus-like particles (VLPs) from fecal samples. The samples process procedure was performed to complete this aim. Please rephrase this sentence.

Resp: This was changed in the new version of the manuscript.

Page 2, Line 79, “The study of VLPs in 250 individuals with gastroenteritis (GI), identified a variety of viruses including GI in humans (RV, Nov or HAdV).” Please remove the word “including GI” in this sentence.

Resp: Done

Page 3, Line 103, “between 18 and 23 years, 4 adults aged 24 to 59 years and 2 elderly > 60 In 5 patients” Please add the period between “elderly > 60” and “In 5 patients”.

Resp: Done

Page 3, Line 105, “It is noteworthy that all patients on admission will have GI symptoms.” The authors need to add the description or definition about clinical symptoms of acute gastroenteritis.

Resp: We have updated this sentence in the revised manuscript.

Page 4, Line 177, “The cleaned reads were first assembled individual DBG assemblers, partitioned assemblers or Mira4” Please rephrase this sentence.

Resp: The NGS description was simplified.

.

Page 4, Line 158-179 The authors conducted two different principle-based assemblers (DBG assemblers and CL assemblers) to harvest the longer genomes. However, the assembly procedure described in this manuscript was chaotic so that the readers could not clearly catch the significant points. We suggest that the authors should simplify the description about assembly procedure.

Resp: We have simplified this description in the revised manuscript

Page 4, Line 195, “92,80% (232/250) were children” The comma symbol was incorrect. Please correct this error at several position throughout this manuscript.  

Resp: Our fault, we have updated all these kinds of mistakes in the text.

Page 5, Line 213-220, The authors implemented two attributive clauses (among which) in this paragraph. The content is right for describing the viral taxonomy, but the grammar structures need to be modified.

Resp: This sentence was modified in the manuscript.

Page 6, Line 236-238, “In these patients, the occurrence of coinfection was observed. being more common between double infection, RV and HAdV in 196, RV and NoV in 173 and NoV and HAdV in 157. triple infection was verified being more frequent among RV, HAdV and NoV in 60,80% (152/250) of patients.” Please correct the grammar mistakes. And the number of co-infection is not consistent with the results of Figure 3. Please carefully check and illustrate the phenomenon.

Resp: This was corrected in the new version of the manuscript.

Page 8, Line 294-295, “Detailed descriptions of the characterization of some of the viruses detected in these samples have been previously reported.” Please rephrase this sentence.

Resp: This sentence was removed because it is redundant.

Page 9, Line 340-341, “This finding indicate that more than one virus can trigger the disease or even contribute to clinical severity.” This study did not summarize the relationship between clinical severity and co-infection and did not perform the statistics. This sentence is not precise, please re-organized.

Resp: The discussion section has been updated.

Page 9, Line 352-353, “Our study indicates that although not related with gastroenteritis enteroviruses were commonly found in association with other pathogenic viruses.” Grammar logic mistake.

Resp: This sentence has been updated in the revised manuscript.

Page 9, Line 357, “Another interesting finding identified in the samples from these patients.” Grammar mistakes.

Resp: This sentence has been deleted.

Page 9, Line 358, “This fact highlights the circulation of new viral types among the population,” The authors found a large number of new virus sequences not assigned to classification, but they thought that these genomic sequences belonged to new viral types of circulation. Although these viral sequences were not assigned to classification, it could be the reason that the viral database lack the according viral taxonomy. We don’t not agree with authors that the circulation of new viral types among the population. If the authors want to illustrate the ideas, they need to perform a systemic and deep studies focusing on these new viral types.

Resp: We agree that this comment was overwhelming, and the sentence has been deleted from this revised version of the manuscript.

Figure 1, “Map showing the location of the states (SP, PA, MA and TO) and municipalities in which the 251 samples” Please modify the grammar mistake. And the dpi of all figures in production need be improved.

Resp: We have changed this in figure 1

Figure 2, The authors declare 251 cases in the legend, resulting in contradiction with previous content. “OUTHER VIRUS HUMAN” The figure contains a word grammar mistake.

Resp: This was correct in this revised version of the manuscript

Figure 3, We suggest that the authors could draw another graphic (single bar plot) to show the number of co-infection within the Figure 3. Because the current graphic plot of graphic cause confusion about co-infection and single infection so that we could not easily identify the number of single infection (e.g., the number of RV).

Resp: We have modified figure 3 accordingly

Table S2. We did not find the Table S2 in the supplementary materials.

Resp: We have uploaded Table S2 and it is available now.

Round 2

Reviewer 2 Report

The authors addressed most of my concerns. 

Reviewer 3 Report

This study investigated the enteric viral composition of 250 individuals with acute gastroenteritis in Brazil. Due to lack of relevant data released in Brazil, the value of this manuscript was significant. 

The authors have carefully revised according to the reviewers' suggestions, and the quality of the paper has been greatly improved.